# PSMA Expression in 122 Treatment Naive Glioma Patients Related to Tumor Metabolism in ^11^C-Methionine PET and Survival

**DOI:** 10.3390/jpm11070624

**Published:** 2021-06-30

**Authors:** Tatjana Traub-Weidinger, Nina Poetsch, Adelheid Woehrer, Eva-Maria Klebermass, Tatjana Bachnik, Matthias Preusser, Mario Mischkulnig, Barbara Kiesel, Georg Widhalm, Markus Mitterhauser, Marcus Hacker, Oskar Koperek

**Affiliations:** 1Division of Nuclear Medicine, Department of Biomedical Imaging and Image-Guided Therapy, Medical University of Vienna, Waehringer Guertel 18–20, 1090 Vienna, Austria; nina.poetsch@meduniwien.ac.at (N.P.); eva-maria.klebermass@meduniwien.ac.at (E.-M.K.); Tatjana_Bachnik@web.de (T.B.); markus.mitterhauser@meduniwien.ac.at (M.M.); marcus.hacker@meduniwien.ac.at (M.H.); 2Division of Neuropathology and Neurochemistry, Department of Neurology, Medical University of Vienna, Waehringer Guertel 18–20, 1090 Vienna, Austria; adelheid.woehrer@meduniwien.ac.at; 3Ludwig-Boltzmann Institute, Nußdorfer Straße 64, 1090 Vienna, Austria; 4Division of Oncology, Department of Internal Medicine I, Medical University of Vienna, Waehringer Guertel 18–20, 1090 Vienna, Austria; matthias.preusser@meduniwien.ac.at; 5Department of Neurosurgery, Medical University of Vienna, Waehringer Guertel 18–20, 1090 Vienna, Austria; Mario.mischkulnig@meduniwien.ac.at (M.M.); barbara.kiesel@meduniwien.ac.at (B.K.); georg.widhalm@meduniwien.ac.at (G.W.); 6Institute of Pathology, Medical University of Vienna, Waehringer Guertel 18–20, 1090 Vienna, Austria; oskar.koperek@meduniwien.ac.at

**Keywords:** glioma, PSMA expression, methionine, PET

## Abstract

Apart from its expression in benign and malignant prostate tissue, prostate specific membrane antigen (PSMA) was shown to be expressed specifically in the neovasculature of solid tumors. For gliomas only little information exists. Therefore, we aimed to correlate PSMA expression in gliomas to tumor metabolism by L-[S-methyl-^11^C]methionine (MET) PET and survival. Therefore, immunohistochemical staining (IHC) for isocitrate dehydrogenase 1-R132H (IDH1-R132H) mutation and PSMA expression was performed on the paraffin embedded tissue samples of 122 treatment-naive glioma patients. The IHC results were then related to the pre-therapeutic semiquantitative MET PET data and patients’ survival. Vascular PSMA expression was observed in 26 of 122 samples and was rather specific for high-grade gliomas ([HGG] 81% of glioblastoma multiforme, 10% of WHO grade III and just 2% of grade II gliomas). Significantly higher amounts of gliomas without verifiable IDH1-R132H mutation showed vascular PSMA expression. Significantly shorter median survival times were seen for patients with vascular PSMA staining in all tumors as well as HGG only. Additionally, significantly higher numbers of PSMA staining vessels were found in tumors with high amino acid metabolic rates. Vascular PSMA expression in gliomas was seen as a high-grade specific feature associated with elevated amino acid metabolism and short survival.

## 1. Introduction

The prostate specific membrane antigen (PSMA) is a type II transmembrane glycoprotein physiologically expressed in the kidney, proximal small intestine, and salivary glands. Moreover, this transmembrane glycoprotein is also known as glutamate carboxypeptidase II (GCPII), as folate hydrolase I (FOLH1) and as N-acetyl-L-aspartyl-L-glutamate peptidase I (NAALADase) in the CNS. Besides its various names PSMA has numerous functions in the body. It is involved in the generation of folate and consecutively in the uptake of folate in the small intestine. In the CNS, NAALADase exhibits neuropeptidase activity by cleaning the neurotransmitter glutamate, an important peptide neurotransmitter and agonist of the glutamate receptor 3, from the neurodipeptide N-acetyl-aspartyl-glutamate (NAAG) at neuronal synapses [1,2]. Moreover, PSMA as a transmembrane protein can be internalized into the cell where glycosylation mediates its proteolytic activity [3]. Internalization may be spontaneous or driven by antibody interaction, which enhances the process threefold [2,4]. Alongside its physiological expression, PSMA expression was first observed to be increased manyfold in prostate cancer cells [5,6]. Consecutively, different radiolabeled PSMA analogues were created and have recently been successfully implemented as a diagnostic imaging tool for patients with advanced prostate cancer with reported sensitivities up to 97% for PSMA compounds labeled with ^68^Gallium (^68^Ga), depending on prostate specific antigen (PSA) serum levels [7,8]. Moreover, recent scientific advances have also led to promising results using PSMA ligands radiolabeled with ^177^Lutetium (^177^Lu) for therapeutic purposes in advanced-stage prostate cancer patients with encouraging results [9,10,11,12]. Besides prostate cancer, PSMA expression was also shown in other tumor entities such as renal cell, urothelial and colon carcinomas [13,14]. In particular, a high PSMA expression in the endothelium of neovasculature of solid tumors but not in normal vessels has been observed [13,14,15,16,17,18]. These conditions permit us to think about a theranostic approach in nuclear medicine for tumors other than prostate cancer.

Gliomas, as brain tumors of a non-metastatic nature, arising from tissue of the central nervous system (CNS), are relatively rare. Nevertheless, approximately 2% of all cancer deaths are glioma-related [19]. Therefore, on the one hand an accurate diagnosis is crucial and on the other hand effective treatment options are needed. Conventional radiologic imaging using CT and MRI has some shortcomings in brain tumor imaging, such as misjudgment of actual tumor extent due to edema or challenging differentiation between pseudoprogression and actual tumor progression.

Nuclear medicine procedures such as the amino acid positron emission tomography (PET) using L-[S-methyl-^11^C]methionine (MET) or O-(2-[^18^F]fluoroethyl)-L-tyrosine (FET) offer additional insights beyond radiological techniques. This information about the biology of gliomas at the time of diagnosis is substantial for differential diagnosis, noninvasive grading, and treatment planning with delineation of tumor extent prior to surgery and radiotherapy. Moreover, it also plays an important role in the post-treatment surveillance and prognostication of these brain tumors [20]. Besides the challenging assessment of the status of brain tumors at an imaging level, the choice of treatment is also demanding. Conventional therapeutic approaches, such as radio- and/or chemotherapy, as well as surgery, are only partly promising. Alternative therapeutic strategies are especially needed to treat malignant gliomas.

There are only limited and heterogeneous data available regarding PSMA expression in gliomas. Recently, positive vascular PSMA staining has been described in a small number of glioblastoma (GBM) with significantly lower PSMA expression in gliomas of lower grade [15,17,18,21]. These observations encourage us to go further into detail for a better understanding of tumor metabolism and behavior and to think about potential new nuclear medicine applications in glioma patients. Therefore, we aimed to analyze PSMA expression by immunohistochemical staining (IHC) in glioma tissues and to relate the observed PSMA expression to tumor metabolism measured by MET PET and survival in a large, treatment-naïve glioma cohort.

## 2. Materials and Methods

### 2.1. Patient Characteristics and Tumor Samples

This retrospective study was approved by the local Ethics Committee. All investigated tumor specimens were acquired from clinical samples obtained by resection or biopsy procedures during the diagnostic workup of 122 glioma patients (67 men, 55 women, mean age: 45, range 18–84 years) diagnosed between 2000 and 2014 without any prior glioma specific therapy. Primary evaluation of all tumor samples was performed prior to the implementation of the current WHO 2016 classification and therefore samples were classified based on the WHO classification 2007 and complimented with additional immunostaining of the isocitrate dehydrogenase 1 R132H (i.e., IDH1-R132H) mutational status as described recently [22]. Immunohistochemistry for Ki67 was performed by an automated slide processing system (Autostainer Plus Link, Dako, Glostrup, Denmark; MIB-1/clone M7240, Dako). The Ki67 proliferation index was determined by counting 500 cells in the most densely stained area and was expressed as percentage (0–100%). A subpopulation was also re-evaluated according to the current WHO 2016 classification. Patients’ characteristics are given in Table 1. Moreover, from each patient, preoperative and pretreatment MET PET data was available. The study population analyzed has already been described in detail in a previous investigation with different purpose, analysis and results [22].

### 2.2. PSMA Immunostaining

Formalin-fixed paraffin imbedded tumor specimens were sectioned at 4 µm intervals and was deparaffinized and rehydrated for further processing. Sections were heated for 64 min at 95 °C in a conventional buffered solution (Ultra Conditioner, Cell Conditioner Nr.1, Ventana Medical Systems/Roche, Tucson, AZ, USA). Indirect IHC for PSMA was performed using a primary antibody (Rabbit Monoclonal, clone: EP192, Epitomics, Abcam, Burlingame, CA, USA) in a 1:50 solution by Epitomics AC-0160. The performance of this antibody for glioma PSMA staining has been validated by Wernicke et al. [18]. A BenchMark^®^ ULTRA-automat (Ventana Medical Systems/Roche, Tucson, AZ, USA) was used for immunohistochemical staining with 3.3′-Diaminobenzidine (DAB). As a secondary antibody an ultraView Universal DAB Detection Kit (Ventana Medical Systems/Roche, Tucson, AZ, USA) was applied. Furthermore, negative controls were obtained by omitting the primary antibody. PSMA vascular immunostaining was assessed according to Birner et al. [23,24]. In a first step, scanned at low magnitude (×40) the immunostained tumor sections were analyzed for the area with the highest density of decorated microvessel cells (MVC; i.e., hot spot). In a second step, MVC were counted in the chosen field by counting all vessels at a total magnification of ×200 within an examination area of 0.25 mm^2^. Each stained lumen was defined as a countable microvessel. The glioma specimens were firstly rated as positive or negative for PSMA IHC. Secondly, the extent of countable decorated endothelial vasculature was assessed by an experienced pathologist, defining stained vascular cell counts per high-power field in a hot spot area as follows: 0 (negative), 1–25, 26–40, >40. Moreover, observed PSMA expression of non-vascular cells was also quantitatively assessed by counting stained cells per high-power field in a hot spot area.

### 2.3. PSMA Immunostaining and Study Population Characteristics

PSMA staining results were related to tumor histology, grading and the IDH1-R132H mutational status (Table 1) as well as patients’ survival. Moreover, tumor to background (i.e., T/N) ratios from preoperatively performed MET PET scans as a read-out for amino acid metabolism were compared with the PSMA immunostaining results. MET PET analysis of this study population was already recently described in detail by Poetsch et al., [22].

### 2.4. MET PET

MET PET was performed using a dedicated full-ring GE Advance PET scanner (General Electric Medical Systems, field of view: 14.875 cm, 35 slices per PET examination with a slice thickness of 4.25 mm). Image acquisition started 20 min after intravenous application of ~740 MBq MET (produced in-house with a radiochemical purity of >97%). Data reconstruction was performed by filtered back projection using a Hanning filter with a cutoff value of 6.2 mm and a 128 × 128 matrix. Image evaluation was performed by 2 experienced nuclear medicine physicians using Hermes software (Gold 3 Hermes Hybrid Viewer, Hermes Medical Solutions, Stockholm, Sweden) manually drawing volumes of interest containing the highest tracer uptake in the tumor and in the contralateral hemisphere for calculating the Tumor/Normal cerebrum (T/N) ratio. Further details see also Poetsch et al. [22].

### 2.5. Statistical Analysis

All applied statistical tests were two-sided and *p*-values <0.05 were considered statistically significant. For statistical analysis, the statistical software SPSS 25.0 for Mac (SPSS 25, SPSS Inc., Chicago, IL, USA) was used. Contingency tables, *t*-tests, Mann–Whitney U-test and chi-square tests were used for descriptive statistical analysis and for analyzing group differences. Survival probabilities were calculated by the product limit method of Kaplan and Meier. Differences between groups were tested using the log-rank test. Due to the low number of PSMA vascular positive cases statistical calculation was performed comparing negative and positive cases irrespective of the quantity of the stained vessels and no further subgroup analysis was performed.

## 3. Results

### 3.1. PSMA Staining

In the entire study cohort, vascular PSMA expression was observed in 26 out of 122 (21% of cases) investigated glioma samples. Distinct differences between high-grade (HGG) and low-grade gliomas (LGG) were seen with positive vessel staining for PSMA in 81% of GBM, 10% of WHO grade III and just 2% of grade II gliomas. In the subgroup of WHO 2016 classified gliomas positive vessel staining was also only seen in high grade tumors (10% anaplastic AC, 12.5% anaplastic ODG and 80% GBM). (Table 2). Moreover, significantly higher PSMA staining vessel-counts were seen in HGG compared to LGG (*p* > 0.001, HGG: 12.8 vessels, LGG: 0.3 vessels). With respect to the IDH1-R132H mutation status, vascular PSMA expression was observed more often in IDH1-R132H wild-type gliomas (WT) as compared to IDH1-R132H mutated gliomas (*p* < 0.001; 35% of WT, 6% of IDH1-R132H mutated, Table 2). Additionally, significantly higher PSMA staining vessel-counts were observed in WT compared to IDH1-R132H mutated gliomas (*p* < 0.001, WT: 13.3 vessels, IDH1-R132H mutated gliomas: 1.4 vessels). An overview of vessel counts in glioma tissue is given in Table 3. Moreover, significantly shorter median survival times as well as higher Ki-67 values were seen for patients with vascular PSMA staining (*p* < 0.001; 11.3 yrs and 1.5 yrs., OR 7.7 and *p* < 0.001; Ki-67: 8% vs. 42%). After adjustment for the IDH1-R132H mutation status, patients showing vascular PSMA expression had significantly shorter survival times in both groups (Figure 1). In addition, survival differed significantly between patients with positive and negative PSMA vessel staining in subgroups of HGG, AC and ODG (*p* < 0.001, OR 5.5; *p* < 0.001, OR 4.8; and *p* = 0.027, OR 6.2). A light non-vascular cell staining was also observed; the amount of non-vascular PSMA staining in the investigated glioma tissue samples did not differ significantly between HGG (mean amount of PSMA staining non-vascular cells: 22) and LGG (20, *p* = 0.26). Furthermore, no differences in non-vascular PSMA expression were seen between AC and ODG (mean amount of PSMA staining cells: cells: 17 and 26, *p* = 0.17), as well as IDH1-R132H mutated and wild type gliomas (mean amount of PSMA staining cells: 18 and 23, *p* = 0.53).

### 3.2. PSMA Staining, Amino Acid Metabolism and Survival

Analyzing amino acid metabolism estimated by T/N ratios of MET PET and PSMA staining, higher vascular PSMA counts were seen in tumors with higher metabolic activity with a T/N ratio cutoff of 2.4 [22] (median 17 vs. 29 PSMA staining vessels, *p* < 0.001). In GBM only non-significant differences in mean T/N ratio values were seen between patients with immunohistochemically verified vascular PSMA expression and those without (*p* = 0.69). Once again using the T/N ratio threshold of 2.4, as recently described [22], for selecting a contingent with high MET uptake, also led to significant survival differences between patients with and without vascular PSMA staining (median survival 2.1 and 7.3 yrs, *p* < 0.001, OR 3.7; Figure 2 and Figure 3). A short overview of patient characteristics in relation to vascular PSMA expression is given in Table 4.

## 4. Discussion

The presented study investigated for the first time a large collective of glioma tissue samples for immunohistochemical PSMA expression with respect to clinicopathological features in terms of survival. PSMA is well known for its expression in prostate tissue. Studies showed PSMA binding in benign prostate epithelium but not in normal vasculature with an increase in the percentage of cellular PSMA staining in prostate cancer with advancing tumor grade [25]. The growing interest in extra-prostatic PSMA expression may lie in the convincing results for diagnostic as well as therapeutic application of labeled PSMA compounds in prostate cancer. ^68^Ga-labeled PSMA tracers are extensively applied clinically for diagnostic purposes in suspected disease recurrence and primary diagnostic management of prostate cancer in study settings. Even though PET imaging using PSMA targeting tracers for other malignancies is currently the subject of research, the PSMA binding profile is different in other solid tumors. Kasoha et al., reported PSMA expression in healthy breast tissue only on normal glandular cells, while tumorous tissue as well as its metastases showed PSMA expression in tumor cells and in tumor associated neovasculature. Moreover, they confirmed higher PSMA expression in neovasculature of distant metastases than in those of primary tumors [26]. These findings are also supported by others, reporting PSMA as being highly and specifically expressed in the tumor neovasculature of ovarian, cervical and endometrial cancer [27].

PSMA expression has also been successfully verified in gliomas [13,16,17]. Comparable to experiences with extra-cranial malignancies, PSMA staining was highly specific for glioma-associated neovasculature. Nomura et al. examined 19 glioma samples of different grades for PSMA expression in comparison with normal brain tissue controls from autopsies. By quantifying the PSMA staining, they found intense staining of the vessels in GBM, moderate staining in grade I tumors and hardly any vessel staining in grade II and III gliomas [17]. Recently, Matsuda et al. also reported similar results [21]. This is in line with our findings that the vast majority of GBM (>80%) showed vascular PSMA expression, while only about 10% of gliomas of lower grades had verifiable vessel staining. Nevertheless, we observed a high number of vessels with PSMA expression (i.e., >40 vessels in only one third of PSMA positive GBM tissue samples), which may be important to help us better understand PSMA tracer uptake behavior in GBM in imaging studies.

Moreover, the quoted studies examined only small glioma populations limited to histopathologic features without clinical correlation. To our knowledge, this is the first description of PSMA expression in gliomas in relation to survival and the IDH1-R132H mutational status, which is one of the key features of the 2016 update of the WHO classification of CNS tumors [28]. As another important point, we found that even after adjustment for IDH-R132H mutational status, the presence of vascular PSMA expression showed an additional negative prognostic value. Further, we found an increased amino-acid metabolism visualized with MET-PET in patients with PSMA positive stained vasculature, both characteristics of aggressive tumor growth, as a possible first attempt to link these two cancer hallmarks that are not interconnected at first glance. In general, the MET uptake shown by PET reflects the increased L-amino transporter activity as well as the enhanced intracellular consumption of amino acids, both characteristic of the malignant processes and aggressiveness of gliomas. We previously showed higher MET uptake to be an indicator for a worse prognosis in glioma patients [22]. The essential amino acid methionine is involved in a plethora of metabolic processes and is a major methyl group donor [29]. However, according to a recent study, increased methionine uptake could have an even more profound impact on cell metabolism than previously thought, as it induces a wide-ranging “metabolic rewiring” towards an anabolic state, e.g., by positively regulating the pentose phosphate pathway [30]. This glucose metabolic pathway is known to not only play a crucial role in tumor cell proliferation and redox defense [31]. but also in migration, proliferation, and tube formation of endothelial cells [32]. Regarding PSMA, the literature indicates that this enzyme also plays a role in facilitating endothelial cell invasion by modulating integrin signaling [33]. Moreover, PSMA fuels signaling via PI3K-Akt, a key metabolic pathway in cancer that generally promotes proliferation, metabolism, cell survival and angiogenesis [34,35]. Lastly, PSMA metabolizes poly-γ-glutamated folates to folates and glutamate, important metabolic substrates that might be taken up excessively by the tumor [36]. While folic acid might prevent cancer development at an early stage, it seems to later promote progression [37]. Importantly, the intracellular paths of methionine and folate are tightly linked, not least because both substances are crucial components of the so-called one-carbon metabolic network, which contributes to a variety of cellular processes, such as the synthesis of nucleotides, polyamines and glutathione for reducing oxidative stress, the latter process also involving the incorporation of glutamate [38]. When these findings are taken together, a connection between vascular PSMA expression and increased MET tumor uptake might not be far-fetched, but seemingly logical.

The first results of small studies using radiolabeled PSMA ligands for diagnostic PET imaging in gliomas are encouraging and show high T/N ratio values [21,39,40,41,42,43]. Moreover, these observations make radiolabeled PSMA compounds candidates for a theranostic application. While studies investigating ^117^Lu-labeled PSMA as theranostic agent for prostate cancer show encouraging results with good response rates and low toxicities [9,44,45,46,47], little is known for high grade gliomas. The first studies aiming at evaluating the clinical implication of PSMA-activated prodrugs (NCT02067156) and PSMA antibody drug conjugates (NCT01856933) for GBM therapy are currently being conducted.

Besides the observed vascular PSMA expression in the vast majority of GBMs and the shorter survival times in patients with, compared to patients without, vascular PSMA expression, vascular PSMA staining was more frequently seen in IDH1-R132H wild-type gliomas, which are commonly known for having a worse prognosis than IDH1-R132H mutated ones [48]. When linking vascular PSMA expression to tumor metabolism and survival, these data seem to reflect previous findings showing that IDH mutation in gliomas leads to a decrease in HIF1A activation and, consequently, to downstream inhibition of hypoxia, as well as vasculo- and angiogenesis related signaling [49]. This link between IDH and PSMA status, might on the one hand, open a new way of looking at glioma characterization based on biomarker evaluation in glioma tissue samples in relation to prognosis and, on the other hand, open up new theranostic perspectives by implementing radioactively labeled PSMA ligands in the therapeutic management of glioma. Despite this, it is necessary to be aware that GBM has a poor survival outcome and patient selection must be considered carefully.

## 5. Conclusions

The presented study shows for the first time a linkage of the tumor neovasculature-specific PSMA expression in glioma tissue with prognostic features such as amino acid tumor metabolism, IDH mutation status and survival. Moreover, although a positive PSMA vessel staining was found to be rather specific in GBM, a high number of PSMA stained vessels was only seen in one third of them, which is important to consider for using radiolabeled PSMA compounds in clinical study settings.

## Figures and Tables

**Figure 1 jpm-11-00624-f001:**
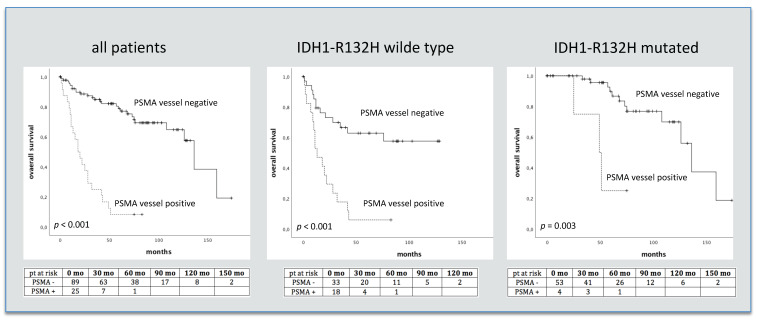
PSMA IHC vessel staining and survival in 122 glioma patients and separated regarding IDH-R132H mutation status. Numbers of patients at risk are given in the tables below. pt = patients, mo = months.

**Figure 2 jpm-11-00624-f002:**
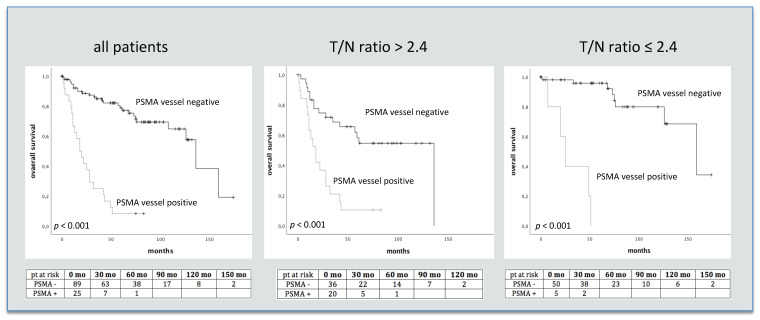
PSMA IHC vessel staining and survival in 122 glioma patients separated by T/N ratio values. Numbers of patients at risk are given in the tables below. pt = patients, mo = months.

**Figure 3 jpm-11-00624-f003:**
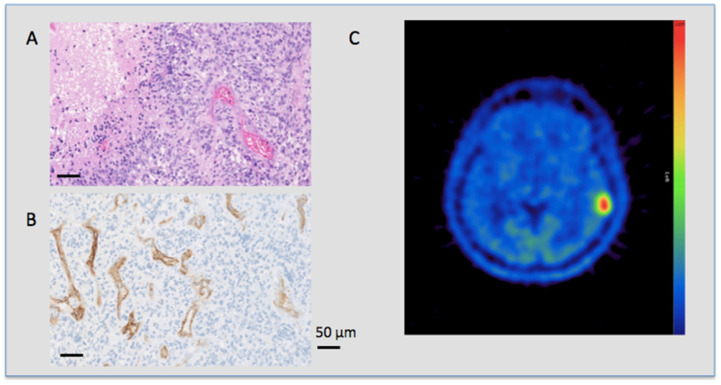
PSMA expression and metabolic tumor activity in a glioblastoma patient. HE (upper left side) and PSMA (lower left side) staining and the corresponding MET PET scan (right side) of a 58 year old patient with an IDH-R132H negative GBM. The patient died 19 months after diagnosis. (**A**) HE staining in 20-fold magnification of a GBM showing a typical crowded, polymorphic cell count and dense vessels. (**B**) PSMA IHC also in 20-fold magnification of the same patient showing an intense reaction on the vascular endothelium with no staining of non-vascular cells. (**C**) MET PET scan of the same patient showing a high focal uptake in the tumor region in the left temporal lobe with a high T/N ratio value of 4.98.

**Table 1 jpm-11-00624-t001:** Patient characteristics.

**Age**	
Mean ± SD	44.5 ± 14.6
Median	44.7
Range	18–83.6
**Sex**	
Male	67
Female	55
**KPS**	
Median	85
Range	20–100
**Histology and Grading**	
LGG/HGG	56/66
Pilocytic astrocytoma	2
Astrocytoma (II/III/GBM)	26/16/26
Oligodendroglioma (II/III)	15/13
Oligoastrocytoma (II/III)	12/11
Ganglioglioma	1
**IDH1-R132H mutation status**	
Mutated	63
Wild-type	53
NA	6

**Table 2 jpm-11-00624-t002:** Results of PSMA staining in glioma tissue samples of 122 patients regarding the IDH1-R132H mutation status, histology and grading (WHO2007) and in 53 patients with WHO 2016 staged.

	PSMA Vessel Staining	
	Positive (n/%)	Negative (n/%)
**WHO Classification 2007**		
**IDH1-R132H mutation status**		
Mutant	4 (6%)	59 (94%)
Wildtype	19 (36%)	34 (64%)
NA	3	3
**Histology and Grading**		
LGG/HGG	1 (2%)/25 (38%)	55 (98%)/41 (62%)
Pilocytic astrocytoma	0 (0%)	2 (100%)
Astrocytoma (II/III/GBM)	1 (4%)/1 (6%)/21 (81%)	25 (96%)/15 (94%)/5 (19%)
Oligodendroglioma (II/III)	0 (0%)/2 (15%)	15 (100%)/11 (85%)
Oligoastrocytoma (II/III)	0 (0%)/1 (9%)	12 (100%)/10 (91%)
Ganglioglioma	0 (0%)	1 (100%)
**WHO classification 2016**		
Diffuse AC, IDH mt	0 (0%)	11 (100%)
Diffuse AC, IDH wt	0 (0%)	3 (100%)
Gemistocytic AC, IDH mt	0 (0%)	1 (100%)
Anaplastic AC, IDH mt	1 (10%)	9 (90%)
Anaplastic AC, IDH wt	0 (0%)	3 (100%)
Diffuse ODG, IDH mt, 1p19q co-del	0(0%)	12 (100%)
Anaplastic ODG, IDH mt, 1p19q co-del	1 (12.5%)	7 (87.5%)
GBM, IDH wt	4 (80%)	1 (20%)

NA = not available, LGG = low grade glioma, HGG =high grade glioma, GBM = glioblastoma multiforme, AC = astrocytoma, ODG = oligodendroglioma, mt = mutation, wt = wild-type, co-del = co-deletion.

**Table 3 jpm-11-00624-t003:** Quantified results of vascular PSMA staining regarding the IDH1-R132H mutation status, histology and grading.

	Negative	1–25 Vessels	26–40 Vessels	> 40 Vessels	Total
**WHO 2007**					
**IDH1-R132H mutation status**					
Mutant	59	2	1	1	63
Wildtype	34	6	7	6	53
**Histology and Grading**					
LGG/HGG	55/41	1/9	0/8	0/8	56/66
Pilocytic astrocytoma	2	0	0	0	2
Astrocytoma (II/III/GBM)	25/15/5	1/1/6	0/0/7	0/0/8	26/16/26
Oligodendroglioma (II/III)	15/11	0/1	0/1	0/0	15/13
Oligoastrocytoma (II/III)	12/10	0/1	0/0	0/0	12/11
Ganglioglioma	1	0	0	0	1
**WHO 2016**					
Anaplastic AC, IDH mt	9	1			10
Anaplastic ODG, IDH mt, 1p19q co-del	7		1		8
GBM, IDH wt	1	1	2	1	5

GBM = Glioblastoma multiforme, AC = astrocytoma, ODG = oligodendroglioma, mt = mutation, wt = wild-type, co-del = co-deletion.

**Table 4 jpm-11-00624-t004:** Overview of patient characteristics with respect to vascular PSMA expression.

	PSMA Vessel Pos	PSMA Vessel Neg	*p*-Value
LGG/HGG	1/25	55/41	
IDH mt/wt	4/19	59/34	
Karnofsky	70	81.1	0.012
Age (yrs)	54.6	41.7	0.251
Mean T/N ratio	4.2	2.5	0.005
Median survival (yrs)	1.5	11.3	0.000
KI-67 (%)	39.8	10.7	0.000

mt = mutation, wt = wild-type, yrs = years.

## Data Availability

Data are available on request from the corresponding author.

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
