# Peer review of "PSMA Expression in 122 Treatment Naive Glioma Patients Related to Tumor Metabolism in ^11^C-Methionine PET and Survival"

_jpm, 2021, doi:10.3390/jpm11070624_

Round 1
Reviewer 1 Report
The study of Traub-Weidinger et al entitled “PSMA expression in 122 treatment naive glioma patients related to tumor metabolism in 11C-methionine PET and survival”, evaluates PSMA expression in primary glioma tissue, correlating PSMA expression to MET PET data and patient survival. In general, the study is of high interest and contains data from a very decent cohort of 122 gliomas of different classifications. Sadly, in its current form it has severe deficits in terms of clarity, methodology, data presentation and over-interpretation of the results.
The points of criticism are described in more detail below.
1) Many methods and analysis types are not described. E.g. how was Ki67 labeling performed? This also accounts for other undescribed methods.
2) Absolute values of e.g. Ki67 positive nuclei were given. At least that is what I suppose, as the authors did not explain it. The number of Ki67 positive signals (and similar factors) depend on the (local) cell density, thus it is necessary to give relative amounts to the total cell number in the field of view/region of interest.
3) There are virtually no images of the analyzed stainings showing representative results, supporting the numbers presented by the authors. This is highly surprising, as this was the main method for data generation. The only one is figure 3, having a low resolution, making it hard to see any details. Additionally, include scale bars in figure 3a/b and a scale (with number) in the heat map of figure 3c. Furthermore, figure 3c can be cropped to a more reasonable size, so that not most of the image contains background.
4) Although authors referred to their previous study when talking about MET PET, they should briefly describe the method and evaluation.
5) Lots of the data generated by the authors is just described in the text without further visualization, making it hard to keep an overview.
6) What was the ratio for the split on the number of stained vascular cells (line 123/124; 0 (negative), 1-25,26-40, >40)? How stable are results against variations?
7) Kaplan and Meier curves should contain the initial population sizes for each sub-population.
8) The reason for including figure 3 remains elusive to this reviewer.
9) Line 234-237: “Further, we were also able to find a correlation between vascular PSMA expression and increased amino-acid metabolism visualized with MET-PET, both characteristics of aggressive tumor growth, as a first attempt to link these two cancer hallmarks that are not interconnected at first glance.” Authors did just find a correlation, but no causation or any additional hint supporting such a notion. Similarly, lines 248-261, while certainly correct, are highly speculative and only very remotely supported by to the data of the authors.
10) In general, the discussion contains too many highly speculative points. Authors should pay attention not to over-interpret their results.
11) The list of references is duplicated.
Author Response
Review Report (Reviewer 1)
The study of Traub-Weidinger et al entitled “PSMA expression in 122 treatment naive glioma patients related to tumor metabolism in 11C-methionine PET and survival”, evaluates PSMA expression in primary glioma tissue, correlating PSMA expression to MET PET data and patient survival. In general, the study is of high interest and contains data from a very decent cohort of 122 gliomas of different classifications. Sadly, in its current form it has severe deficits in terms of clarity, methodology, data presentation and over-interpretation of the results.
The points of criticism are described in more detail below.
We thank the reviewer for the insightful suggestions and comments. We tried to answer all questions and to incorporate all suggestions into the manuscript. Please find below our answers and corrections point to point.
1) Many methods and analysis types are not described. E.g. how was Ki67 labeling performed? This also accounts for other undescribed methods.
We understand to describe the analyzing methods more precisely. Therefore, we referenced the immunostaining of IDH1-R132H and added a short description of Ki labeling in the method section (see 2.1, line103-107).
2) Absolute values of e.g. Ki67 positive nuclei were given. At least that is what I suppose, as the authors did not explain it. The number of Ki67 positive signals (and similar factors) depend on the (local) cell density, thus it is necessary to give relative amounts to the total cell number in the field of view/region of interest.
Please see also Point 1. We added a short description of Ki labeling and analyzing in the method section.
3) There are virtually no images of the analyzed stainings showing representative results, supporting the numbers presented by the authors. This is highly surprising, as this was the main method for data generation. The only one is figure 3, having a low resolution, making it hard to see any details. Additionally, include scale bars in figure 3a/b and a scale (with number) in the heat map of figure 3c. Furthermore, figure 3c can be cropped to a more reasonable size, so that not most of the image contains background.
We thank for the comments and adapted figure 3a and b in regard to better resolution (20-fold magnification) and included scale bars as suggested. Moreover, we cropped figure 3c with less background and added a heat map (line 218).
4) Although authors referred to their previous study when talking about MET PET, they should briefly describe the method and evaluation.
Accordingly to the proposal we described briefly the method and evaluation of MET PET in the method section adjusting a further subsection (2.4, line 142-152).
5) Lots of the data generated by the authors is just described in the text without further visualization, making it hard to keep an overview.
As suggested by the reviewer we adapted our manuscript for better keeping the data more in overview and added a Table (Table 4) of patients’ characteristics with respect to vascular PSMA expression to the results section (see 3.2., line 215-216). Moreover, as also mentioned under point 7 we modified the Kaplan Meier curves (Figure 1,2, line 194 and line 217) adding patients at risk and added a table including main results of the analysis.
6) What was the ratio for the split on the number of stained vascular cells (line 123/124; 0 (negative), 1-25,26-40, >40)? How stable are results against variations?
Thank you for your comment. Due to the low number of PSMA vascular positive cases statistical calculation was performed comparing negative and positive cases irrespective of the quantity of the stained vessels (we adjusted this also in the statistics part line 161-165). Additionally the number of stained vessels (numeric data) correlated with the grade of the tumor as well as with IDH-mutation status (see manuscript results 3.1 PSMA Immunostaining). Thus, for demonstrative purposes only we subtyped the positive cases in the table 3 in three evenly distributed subgroups, namely 1-25 (low), 25-40 (moderate), >40 (prominent).
7) Kaplan and Meier curves should contain the initial population sizes for each sub-population.
As suggested we adapted the Kaplan Meier curves (Figues 1,2) adding patients at risk for all subgroups according to IDH Status and metabolic tumor activity. We hope this contributes to a better and more detailed understanding.
8) The reason for including figure 3 remains elusive to this reviewer.
For better understanding and illustration we adapted the figure 3 (line 218) accordingly to the reviewer´s suggestions (see also point 3) and also supplemented a headline.
9) Line 234-237: “Further, we were also able to find a correlation between vascular PSMA expression and increased amino-acid metabolism visualized with MET-PET, both characteristics of aggressive tumor growth, as a first attempt to link these two cancer hallmarks that are not interconnected at first glance.” Authors did just find a correlation, but no causation or any additional hint supporting such a notion. Similarly, lines 248-261, while certainly correct, are highly speculative and only very remotely supported by to the data of the authors.
Please see our comments under point 10.
10) In general, the discussion contains too many highly speculative points. Authors should pay attention not to over-interpret their results.
We thank the reviewer for his advice and adapted the discussion. For the first time we tried to evaluate vascular PSMA expression together with Methionine PET bringing together two prognostic feature in gliomas, one histopathological and one imaging related. We did not intend to over interpret our results but nevertheless tried to put our results in context to the already published data regarding metabolic tumor processes. Therefore we adapted the discussion section accordingly (line 273-279).
11) The list of references is duplicated.
Thank you for your thoughtful review of our work. We have corrected the references.
Reviewer 2 Report
This is an interesting paper. However, while PMSA vascular expression in GBMs suggested shorter median survival, this is unlikely to influence treat modification. GBMs have a poor survival outcome. I think this should be addressed in the discussion
The sections on results are satisfactory
Author Response
Review Report (Reviewer 2)
This is an interesting paper. However, while PMSA vascular expression in GBMs suggested shorter median survival, this is unlikely to influence treat modification. GBMs have a poor survival outcome. I think this should be addressed in the discussion.
We thank the reviewer for the comments. We adapted according to the reviewer´s considerations in the discussion section (line 236-238).
The sections on results are satisfactory.
Reviewer 3 Report
The paper is very original in the description of correlation among PSMA expression and IDH R132H mutation / metabolism detected in MET-PET / survival. Do you think that in the future PSMA should be performed routinely in the pathological report? Of course new data will be required, and your study is only retrospective, but it's very interesting for now.
At line 98, why do you use WHO 2007 classification? (maybe you can specify that the analysis was done before 2016).
Author Response
Review Report (Reviewer 3)
The paper is very original in the description of correlation among PSMA expression and IDH R132H mutation / metabolism detected in MET-PET / survival. Do you think that in the future PSMA should be performed routinely in the pathological report? Of course new data will be required, and your study is only retrospective, but it's very interesting for now.
We thank the reviewer for this interesting suggestion. We added a short sentence to the discussion section (see line 333-335).
At line 98, why do you use WHO 2007 classification? (maybe you can specify that the analysis was done before 2016).
We thank the reviewer for this advice and have supplemented it in the method section (line 101-109) according to his suggestion to specify that the main analysis was done before 2016.
Round 2
Reviewer 1 Report
The authors covered all questions raised.